# Underlying Diagnosis of Food Bolus Obstruction Acute Admissions to Otorhinolaryngology: A Shift to Provide the Best Care as per the Current Guidelines

**DOI:** 10.3390/medicina61061047

**Published:** 2025-06-06

**Authors:** Rasads Misirovs, Anna Kamusella, Michael Miller, Samit Majumdar

**Affiliations:** 1Department of Otorhinolaryngology, NHS Lothian, Edinburgh EH16 4SA, UK; 2Department of Doctoral Studies, Riga Stradins University, LV-1007 Riga, Latvia; 3Department of Psychiatry, Edinburgh, NHS Lothian, Edinburgh EH10 5HF, UK; 4Department of Gastroenterology, Ninewells Hospital & Medical School, NHS Tayside, Dundee DD1 9SY, UK; 5Department of Otorhinolaryngology, Ninewells Hospital & Medical School, NHS Tayside, Dundee DD1 9SY, UK

**Keywords:** oesophagogastroduodenoscopy, dysphagia, swallowing disorder, eosinophilic esophagitis, biopsy

## Abstract

*Background and Objectives:* In the United Kingdom, some patients with food bolus obstruction (FBO) are admitted under the care of ear, nose, and throat (ENT) doctors. In the literature, eosinophilic oesophagitis (EoE) is the most common cause of FBO. We analysed ENT FBO admissions and interventions used in our hospital to investigate for EoE. *Materials and Methods:* This paper details a retrospective study of adult FBO admissions to an ENT ward from January 2016 to December 2019 at a single centre. *Results:* In total, 120 patients were admitted. Half of the patients required instrumentation to resolve the obstruction—31% underwent rigid oesophagoscopy (RO) and 69% oesophagogastroduodenoscopy (OGD). Biopsies were taken during 48% of inpatient OGDs and 5% of ROs. 48% had a histopathological diagnosis of EoE. There was no mention of a specific number of eosinophils per high-power field in 53% of EoE pathology reports. Potentially, some patients were EoE-negative due to an inadequate number of biopsies taken—71% of patients had an insufficient number of biopsies to exclude EoE. A total of 56% of all patients with FBO did not undergo inpatient or outpatient OGDs with biopsies. *Conclusions:* Biopsies were not taken in all FBO patients undergoing oesophagoscopy, leaving EoE underdiagnosed. Follow-up arrangements were often suboptimal to exclude EoE.

## 1. Introduction

Food, as well as any other material, ingested wilfully or accidentally, can become stuck in the oesophagus, resulting in oesophageal obstruction [1]. Oesophageal food bolus obstruction (FBO) is a common problem, with an estimated annual incidence of 13/100,000 people [2]. Many such cases of obstructions resolve spontaneously. Others may present acutely to hospital emergency departments and subsequently be referred to ear, nose, and throat (ENT) surgeons or gastroenterologists (GIs) for further management [1]. In our health board, patients with soft FBO are admitted to ENT if the suspected obstruction is above the sternal notch; if below the sternal notch, they are admitted under GIs/medics or general surgeons. Ntuli et al. reported that at our institution, between 2008 and 2014, 46.8% of 310 acute FBO admissions were managed by ENT, 43.5% by general surgery, and 9.7% by general medicine [3]. The practice varies between centres in the United Kingdom (UK) depending on a few factors: (1) whether ENT or GI doctors/surgeons are in the same hospital with accident and emergency departments, (2) access to flexible oesophagogastroduodenoscopy (OGD) performed by GI doctors or upper GI surgeons in-hours and out-of-hours, and (3) arrangements between ENT, GI doctors, and upper GI surgeons regarding which team patients with FBO are admitted under.

Soft FBO may resolve with ‘conservative’ strategies. Oesophagoscopy is considered the definitive management for persistent cases of food bolus impaction [1]. The European Society of Gastrointestinal Endoscopy (ESGE) and American Society for Gastrointestinal Endoscopy (ASGEE) recommend definitive management of oesophageal FBO within 24 h to prevent complications from delayed endoscopic interventions. However, the ESGE recommends curative intervention in the first 6 h for complete oesophageal obstruction [1,2]. The inability to swallow one’s own saliva and any liquids is suspicious for complete oesophageal obstruction [2].

The advised method of intervention to retrieve a food bolus by the ESGE and the ASGE is flexible endoscopy (primarily OGD, but transnasal oesophagoscopy (TNO) can be used as well), but the ASGE does acknowledge the potential usefulness of rigid oesophagoscopy in higher foreign bodies impacted at the hypopharynx or cricothyroid [4]. The use of flexible or rigid endoscopes carries potentially serious complications [1].

A diagnostic work-up after the removal of the food bolus is recommended to detect any underlying disease, as a lack of appropriate follow-up has been shown to be a predictor for recurrent FBO [2]. The most common predisposing pathology of FBO is eosinophilic oesophagitis (EoE) [3,5]. It is important to take an appropriate number of biopsies from several levels of the oesophagus to diagnose or exclude EoE. Several papers have suggested taking six or more biopsies from three levels of the oesophagus [6]. Biopsy reports are critical diagnostic criteria for EoE, specifically the number of eosinophils per high-power field (hpf). EoE is likely if the eosinophil count per hpf is above 15 hpf [5].

We analysed all admissions for acute FBO to our unit and assessed the type of interventions used and the ability of our processes to diagnose or exclude EoE as the cause of FBO. As EoE is the most common cause of FBO, we intended to see whether appropriate inpatient or outpatient biopsies of the oesophagus were performed to diagnose or exclude EoE. Our study highlights the gap between the current practice and the European guidelines on investigating EoE as the most common cause of FBO [6]. We focused only on ENT-managed FBO, not on GI-managed FBO, as we wanted to assess whether we ENT surgeons investigate our FBO patients well enough according to European guidelines on a condition we deal with rarely in the outpatient setting.

## 2. Materials and Methods

### 2.1. Study Design

The STROBE checklist was followed for reporting the findings of this study. This is a retrospective study of adult (≥16 years of age) FBO admissions to the ENT Department in Ninewells Hospital in Dundee, NHS Tayside, Scotland, between January 2016 and December 2019. Patients with FBO were identified through the ENT inpatient ward admission diary. If the same patient had more than one admission during the study period, only the index admission was included in the data analysis. Electronic and paper case notes were accessed. Appropriate clinical governance approvals were granted. Cases of only soft FBO admissions were included in the study; therefore, patients with sharp food objects, such as fishbone or chicken bone, and non-organic foreign bodies were excluded. No standard local protocols were in place specifically, other than already available European guidelines on diagnosing and managing EoE [6].

### 2.2. Limitations

All NHS Tayside acute admissions with FBO are admitted to Ninewells Hospital. Patients living in the northern part of the NHS Fife area are acutely admitted to Ninewells Hospital, but most of the time, outpatient management of these patients is carried out in Victoria Hospital in Kirkcaldy, NHS Fife; therefore, information on outpatient follow-ups of these patients is not available to NHS Tayside staff. We suspect some of this information on further acute FBO admissions and/or outpatient follow-ups of NHS Fife patients has not been available for our study. Some patients moved outside of the NHS Tayside area since acute admission with FBO to Ninewells Hospital. Overall, we do not have complete follow-up information on 20 such patients.

## 3. Results

### 3.1. Demographics

A total of 120 FBO admissions was identified: 82 (68%) were males and 38 (32%) were females. The median age of males was 54.5 (range 16–94) years, while that of females was 69.5 (range 16–92) years, which was statistically significant (*p* = 0.008).

In 87%, the type of food bolus was animal-based, in 8%, it was plant-based, and in the remaining 5%, animal-based food could not be excluded. Of all animal-based food boluses, chicken was the cause of FBO in 36% of admissions, steak—22%, beef—14%, pork—6%, and lamb—5%. The rest was fish, duck, processed food, such as sausages, mince, meatballs, and ham, and unspecified meat; each of these aforementioned food groups was the cause of FBO in less than 5%.

A trend of more admissions during the summer season and winter festive period was noted (Figure 1).

Several elderly patients were admitted around the time of their birthdays. Admissions during the days of the week were more common on weekends and Mondays (Figure 2).

None of the patients had airway compromise.

### 3.2. Interventions

Interestingly, in nearly half (49% or 59/120) of FBO admissions, the obstruction resolved spontaneously within the 24 h observation period. A total of 51% (61/120) of admissions required intervention, rigid oesophagoscopy (RO) or OGD. In 67% (41/61), OGD was the intervention, and in 31% (19/61), it was RO. In one FBO, combined OGD and RO were performed due to cervical osteophytes preventing safe instrumentation with RO.

Two complications of oesophageal tear were recorded, one in the OGD and one in the RO group, leading to a 2.4% and 5.3% complication rate in each group, respectively.

During inpatient OGD, biopsies were taken in 48% (20/42) of cases, while during RO, biopsies were taken in 5% (1/19). Biopsies of inpatient OGD revealed EoE in 50% (10/20) (Figure 3).

None of the patients had a previously known diagnosis of EoE.

### 3.3. Follow-Up

A total of 54% (54/100) of patients in which FBO resolved spontaneously or did not undergo OGD with biopsies during inpatient stay had no follow-up arranged. Outpatient OGD was planned in 16% (16/100) but did not take place in four of the patients for unknown reasons. TNO was planned in 4% (4/100) but did not take place in two of the patients for unknown reasons. An outpatient barium swallow was arranged in 11% (11/100). An outpatient ENT clinic review without OGD/TNO was arranged for 5% (5/100) patients. In 4% (4/100) of the patients, general practitioners (GPs) were advised to request outpatient OGD or barium swallows.

In 92% of outpatient OGDs (11/12), biopsies were taken. In 0% (0/2) of TNO cases, biopsies were taken. The most common diagnosis of outpatient OGD biopsies was EoE in 46% (5/11); in one patient, the diagnosis was adenocarcinoma (Figure 4).

When biopsies were performed during the inpatient and outpatient OGDs, in 71% (22/31), an insufficient number of oesophageal biopsies was taken (too few and/or from less than three levels of the oesophagus); therefore, only 29% of OGD biopsies were taken appropriately to diagnose or to exclude EoE.

In 53% of patients newly diagnosed with EoE, a specific number of eosinophils was not mentioned in their biopsy reports. Instead, pathologists used words such as ‘mild/marked/predominant/dense infiltration of eosinophils’ or similar.

In summary, when oesophageal biopsies were taken during the inpatient and outpatient OGDs, EoE was diagnosed in 48% (15/31) of patients. The mean age of these patients with newly diagnosed EoE was 28 years (range 15–60). The male to female ratio was 1.8:1. Animal-based food was the food bolus type in 100% of these EoE patients.

## 4. Discussion

This study has clearly shown that the majority of patients were under-investigated to rule out or diagnose EoE, the most common cause of FBO. When OGD was performed, only 29% had the recommended six biopsies taken. Despite this deficiency, our study still found 15 patients with EoE, but many more were likely missed due to failure to take adequate biopsies in all cases.

Males tend to be admitted with FBO more commonly compared to females [3,5]. This could be due to underlying pathology, such as EoE, which has been reported to be more common in males, and/or diet habits, such as higher quantities of meat being consumed by males [7,8]. In our cohort, males were admitted more commonly and were younger than females.

More admissions during the summer season and winter months may be explained due to dietary changes, including the increased consumption of barbeque meat during summer months and roasted meat during winter festive months. Similarly, increased admissions over weekends and Mondays may be associated with the higher consumption of meat when dining out. Previously, reports have suggested that increased FBO during summer months is due to increased levels of aeroallergens [9].

In our study, almost half of FBO cases resolved spontaneously, which is similar to a case series reported by Tsikoudas et al.—54% of 37 patients with symptoms of FBO for longer than 24 h found that the obstruction resolved spontaneously [10].

Different parenteral treatments may be given to relax different levels of oesophageal muscles such as benzodiazepines, glucagon, calcium channel blockers, and hyoscine butylbromide, the latter being the one we use in our hospital; nevertheless, we did not investigate the therapeutic effects of parenteral treatment for FBO. As there is no good evidence for any of these muscle relaxants in FBO, it might just be a UK-based ENT preference [11,12].

Instrumenting the oesophagus to resolve FBO can result in complications. In our cohort, one patient instrumented with RO and one with OGD had oesophageal tears, showing a two-fold higher complication rate with RO. Similarly, Straumann et al. reported a 20% (2/10) incidence of oesophageal tears with RO and 0% (0/124) with flexible endoscopy when removing a food bolus obstructing the oesophagus specifically in EoE patients [13]. Nevertheless, a meta-analysis by Ferrari et al. revealed no statistical significance of complication rates between flexible or rigid oesophagoscopy [14].

A general anaesthetic is required to perform all ROs, as well as during OGD/TNO if patients are not cooperative or if there is a high risk of aspiration [2]. Although our hospital follows guidelines on which speciality to refer the patients with FBO to, depending on whether it is above or below the sternal notch, the area of discomfort in the neck or chest often does not correlate with the site of obstruction [2,4]. In our ENT Department, the choice of intervention was decided based on the patients pointing out where the food was stuck. If they pointed in the neck and emergency theatres were available, then we performed RO. But if they pointed behind the sternum or emergency theatres were not available for RO within 8 h, then OGD was organised. Occasionally, a combined approach with OGD and RO might be required. In our cohort, one patient required a combined approach due to cervical osteophytes.

In our cohort, biopsies were taken in 48% (20/42) of inpatient OGDs, which is something that should be focused on in the future—taking biopsies in close to 100% of OGDs when performed for the removal of a soft food bolus. In a small group of our patients, OGDs with biopsies were arranged as a follow-up procedure if they were not performed during the inpatient stay, but 25% of our patients did not attend the arranged follow-up OGD; therefore, the aim should be to take biopsies during the index OGD. If a concern is prolonging the procedure due to taking biopsies, therefore making the procedure intolerable for patients, the operating endoscopists should highly consider performing the OGD under general anaesthetic.

Biopsies of the upper, middle, and lower oesophagus can be performed safely with OGD or TNO, both of which are flexible scopes. In our cohort, EoE was the most common diagnosis from biopsies taken during OGD, at 48% (15/31), with adenocarcinoma at 3% (1/31). A biopsy report of a ‘normal oesophagus’ cannot be considered unless six or more biopsies from three levels of the oesophagus have been taken [12].

As the commonest underlying oesophageal pathology presenting with FBO is EoE, these patients should be managed by GI doctors who can perform OGD to remove food boluses and take oesophageal biopsies to exclude EoE [5]. Since the results of this study were presented at multiple meetings, all FBO patients presenting at our hospital are admitted under GI doctors, as they are ultimately involved in looking after EoE patients. If in some hospitals, due to logistical reasons, FBO admissions are managed by ENT, then outpatient OGD or TNO with biopsies should be arranged for all patients as a follow-up investigation to look for the underlying cause of FBO. ENT should be involved in FBO cases presenting with an airway compromise or when the FBO is high in the oesophagus, preventing the OGD from advancing into the oesophagus.

In our cohort, 54% of patients (54/100) did not have a follow-up arranged. In 44% (20/46) of patients, appropriate follow-up with OGD or TNO was arranged, although the appointments did not take place in 30% (6/20) for unknown reasons (such as cancellation by patients or administrative error). Our findings highlight that the diagnosis of EoE was not considered in the majority of patients admitted to our unit with FBO. This has already been discussed in a systematic review by Bahgat et al. in 2015, but the practice has not improved since then, and further efforts to change the standards of investigation should be undertaken [15]. EoE is becoming a common disease and the number of patients with EoE as the cause of their FBO will rise as the disease itself becomes commoner [16].

## 5. Conclusions

Our retrospective study revealed that the focus of FBO admissions under the care of ENT was to resolve acute oesophageal obstruction. There was less focus on investigating the underlying oesophageal pathology. Patients with FBO should ideally be admitted under the care of GI doctors or upper GI surgeons, as they can perform OGD to resolve acute FBO and can examine the entire length of the oesophagus and take the appropriate number of biopsies. When patients are admitted under ENT care, ENT surgeons should be aware of EoE being the most common oesophageal pathology predisposing patients to FBO; therefore, patients should be referred for OGD or TNO with biopsies.

## Figures and Tables

**Figure 1 medicina-61-01047-f001:**
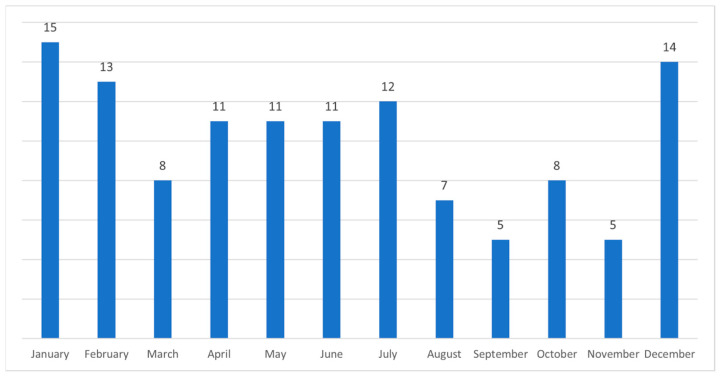
A number of admissions of patients with food bolus obstruction each month of a year.

**Figure 2 medicina-61-01047-f002:**
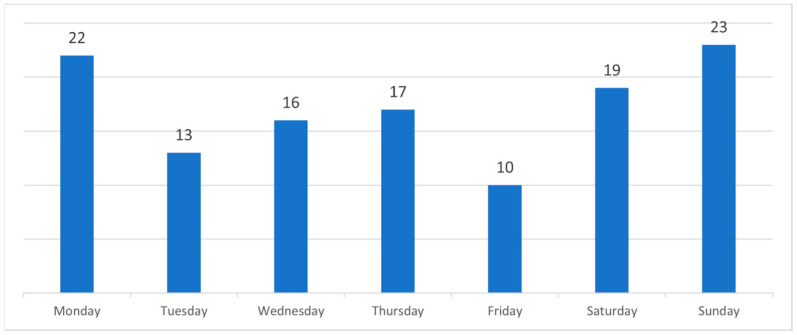
A number of admissions of patients with food bolus obstruction each day of a week.

**Figure 3 medicina-61-01047-f003:**
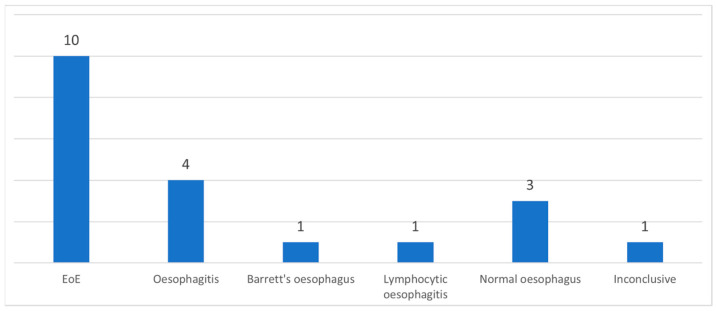
A number of patients with a diagnosis of inpatient OGD biopsies. EoE—eosinophilic oesophagitis.

**Figure 4 medicina-61-01047-f004:**
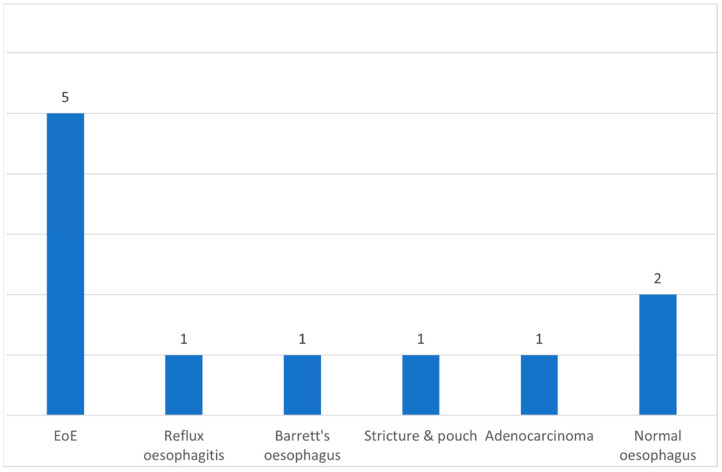
A number of patients with a diagnosis of outpatient OGD biopsies. EoE–eosinophilic oesophagitis.

## Data Availability

Data are unavailable due to NHS Tayside privacy concerns, but, if required, the corresponding author will contact the NHS Tayside Information Governance Department with a request to share anonymized data.

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
