# Peer review of "Underlying Diagnosis of Food Bolus Obstruction Acute Admissions to Otorhinolaryngology: A Shift to Provide the Best Care as per the Current Guidelines"

_medicina, 2025, doi:10.3390/medicina61061047_

Round 1
Reviewer 1 Report
Comments and Suggestions for Authors
1. I am interested, you have stated that during OGD in 48% patients the biopsy was taken and during RO only in 5%. Could you explain the reason for the biopsy? Was it just to prove EoE or was there a visible tumor or any kind of mucosa change?
2. Were there any patients with stenosis of oesophagus?
3. Do you recommend a routine biopsy of oesophagus in each patient who doesn't get spontaneous resolution of obstruction with food bolus?
Author Response
Many thanks for your comments and questions. We appreciate your time reading our manuscript. We hope we have addressed them all here below.
Comment 1: I am interested, you have stated that during OGD in 48% patients the biopsy was taken and during RO only in 5%. Could you explain the reason for the biopsy? Was it just to prove EoE or was there a visible tumor or any kind of mucosa change?
Response 1: In patients with food bolus obstruction (FBO) biopsies should be taken in all patients during the OGD as per guidelines to exclude EoE which is the most common underlying cause for FBO (reference 2 and 6). The biopsies should be taken from the upper, middle and lower levels of the oesophagus which can only be done safely during OGD. Oesophageal mucosa often is macroscopically normal in the early presentation of EoE, therefore biopsies should be taken during OGD in all patients with FBO. We highlight that the biopsies were not taken in more than half of the performed OGD. During the RO biopsies were taken only in 5% which was done due to the abnormal appearance of oesophageal mucosa to primarily exclude malignancy. It is impossible to safely take biopsies with RO from the lower level of the oesophagus therefore we advocate the need to do OGD rather than RO for patients with FBO to exclude EoE.
Comment 2: Were there any patients with stenosis of oesophagus?
Response 2: Oesophageal stenosis is one of the causes of FBO and one of the complications of EoE but none of the patients in our study had stenosis.
Comment 3: Do you recommend a routine biopsy of oesophagus in each patient who doesn't get spontaneous resolution of obstruction with food bolus?
Response 3: Yes, we recommend OGD with biopsies as per the guidelines for all patients with FBO that does not resolve spontaneously. And even if it does resolve spontaneously but they experience recurrent FBO they should undergo outpatient OGD with biopsies to exclude EoE.
Reviewer 2 Report
Comments and Suggestions for Authors
Dear Editor,
The authors examined ENT FBO admissions in their hospital and the interventions used to investigate EoE. 120 cases were retrospectively evaluated. Biopsies were obtained during 48% of inpatient OGDs and 5% of ROs. 48% had a histopathologic diagnosis of EoE. In the study, it was reported that biopsies were not taken from all FBO patients who underwent esophagoscopy and the diagnosis of EoE was insufficient. Unfortunately, this study only includes the data of one clinic in foreign body ingestion and does not provide any new information to the literature.
Sincerely
Author Response
Many thanks for your comments. We appreciate your time reading our manuscript. We hope we have addressed them here below.
Comment 1: The authors examined ENT FBO admissions in their hospital and the interventions used to investigate EoE. 120 cases were retrospectively evaluated. Biopsies were obtained during 48% of inpatient OGDs and 5% of ROs. 48% had a histopathologic diagnosis of EoE. In the study, it was reported that biopsies were not taken from all FBO patients who underwent esophagoscopy and the diagnosis of EoE was insufficient. Unfortunately, this study only includes the data of one clinic in foreign body ingestion and does not provide any new information to the literature.
Response 1: This is a single-centre retrospective study where we report the investigations of patients presenting with food bolus obstruction (FBO) and not being investigated well enough to exclude eosinophilic oesophagitis (EoE) which is the single most common underlying pathology in patients with FBO. We hope we show the importance of not only therapeutic management of FBO but also diagnostic management with multiple biopsies of oesophagus during oesophagogastroduodenoscopy (OGD) which is the way to diagnose or exclude EoE. We have shown the importance of OGD not only during inpatient management but also during outpatient follow-up as the most common diagnosis was EoE in outpatient biopsies and even one case of adenocarcinoma. We have highlighted the importance of pathologists required to mention specific number of eosinophils per high power field as it was not mentioned in over half of the patients. The aim of this manuscript is to spread the word to other ENT departments admitting patients with FBO that diagnosing the underlying cause is paramount to address the future FBO and start the treatment of EoE if it has been diagnosed. Anecdotally across Scotland and the United Kingdom ENT surgeons have been focusing more on therapeutic management of resolving FBO but not excluding or proving EoE with diagnostic OGD. We have referenced previously published literature in the introduction and discussion sections.
If there is anything more specific you would like us to address, we will do it as best as we can. Many thanks.
Reviewer 3 Report
Comments and Suggestions for Authors
Manuscript Title: Underlying diagnosis of food bolus obstruction acute admissions to otorhinolaryngology: a shift to provide the best care as per the current guidelines
This is an interesting and relevant retrospective study that evaluates how food bolus obstruction (FBO) admissions are handled by ENT services in a single UK center, with a focus on the detection of eosinophilic oesophagitis (EoE) as an underlying cause. The manuscript is informative and highlights a potential gap in clinical practice. The study is methodologically sound and well-written. Some minor revisions are needed to improve clarity and completeness.
Major strengths:
- Focus on a relevant clinical problem: underdiagnosis of EoE in patients with FBO.
- Clear data presentation, supported by figures and tables.
- Discussion is well-referenced and contextualized with international guidelines.
Minor revisions needed:
- Introduction:
- Specify more clearly the gap in current practice the study intends to highlight (e.g., failure to follow ESGE/ASGE guidelines regarding biopsies and follow-up).
- Consider briefly expanding the rationale for focusing on ENT-managed FBOs instead of GI-managed ones.
- Methods:
- Clarify if any standard local protocols were in place during the study period regarding biopsy taking or follow-up planning.
- Justify the threshold of ≥6 biopsies from three levels as a standard – perhaps briefly citing the relevant guidelines.
- Results:
- A table summarizing key patient subgroups (e.g., inpatient OGD with biopsy, without biopsy, outpatient OGD, no follow-up) would improve clarity.
- Provide the exact number of patients who received a diagnosis of EoE overall, combining inpatient and outpatient cases.
- Discussion:
- The discussion on complication rates of rigid vs. flexible endoscopy is valuable. Consider briefly explaining how decisions are made locally regarding the choice between OGD and RO.
- Strengthen the practical take-home message: what changes in ENT protocols or interdisciplinary collaboration could improve future EoE detection?
- Conclusion:
- Make the conclusions more direct and action-oriented (e.g., “ENT services should adopt protocols to ensure all FBO cases undergo appropriate follow-up to exclude EoE”).
Comments on the Quality of English Language
The manuscript is well-written and clear. Minor improvements in sentence structure and conciseness are recommended, particularly in the introduction and discussion sections. A language editing service is not required.
Author Response
Many thanks for appreciating the importance of this manuscript in highlighting the underdiagnosis of EoE in patients with FBO presenting to ENT. We read with pleasure your constructive comments and questions. We appreciate your time reading our manuscript and going through the revised version again. We hope we have addressed all your queries here below and we have highlighted the changes made in the manuscript in yellow.
This is an interesting and relevant retrospective study that evaluates how food bolus obstruction (FBO) admissions are handled by ENT services in a single UK center, with a focus on the detection of eosinophilic oesophagitis (EoE) as an underlying cause. The manuscript is informative and highlights a potential gap in clinical practice. The study is methodologically sound and well-written. Some minor revisions are needed to improve clarity and completeness.
Comment 1:
Introduction
- Specify more clearly the gap in current practice the study intends to highlight (e.g., failure to follow ESGE/ASGE guidelines regarding biopsies and follow-up).
- Consider briefly expanding the rationale for focusing on ENT-managed FBOs instead of GI-managed ones.
Response 1:
- Now we have added a sentence (the penultimate sentence in the Introduction section) highlighting the gap in the current practice and the European guidelines on investigating EoE in patients with FBO (reference No. 6).
- In the last sentence of the Introduction section we have briefly explained why we have focused only on ENT-managed FBOs.
Comment 2:
Methods:
- Clarify if any standard local protocols were in place during the study period regarding biopsy taking or follow-up planning.
- Justify the threshold of ≥6 biopsies from three levels as a standard – perhaps briefly citing the relevant guidelines.
Response 2:
- We have added a last sentence in the Methods 2.1. section that no specific local protocols were in place regarding taking biopsies or follow-up planning other than already available European guidelines for diagnosing and managing EoE.
- European guidelines on diagnosing and managing EoE (reference No. 6) instructs to take 6 or more biopsies to exclude or diagnose EoE (see line 68 in the Introduction section where we reference the guidelines).
Comment 3:
Results:
- A table summarizing key patient subgroups (e.g., inpatient OGD with biopsy, without biopsy, outpatient OGD, no follow-up) would improve clarity.
- Provide the exact number of patients who received a diagnosis of EoE overall, combining inpatient and outpatient cases.
Response 3:
- We appreciate your suggestion of a table summarizing key patient subgroups. As we have already broken down the Results section into Demographics, Interventions and Follow-up and have provided the details of the data analysis in these subsections we feel adding a summary table of the results might lead to duplication of the Results text and the suggested Table. Nevertheless, we are open to your suggestions on what to and what not to include in the Table to prevent duplication. We hope this is reasonable.
- We have added a sentence combining overall newly diagnosed EoE. It is the first sentence of the last paragraph in the Results section. We amended the second sentence of this paragraph according to the added sentence prior to it.
Comment 4:
Discussion:
- The discussion on complication rates of rigid vs. flexible endoscopy is valuable. Consider briefly explaining how decisions are made locally regarding the choice between OGD and RO.
- Strengthen the practical take-home message: what changes in ENT protocols or interdisciplinary collaboration could improve future EoE detection?
Response 4:
- We have added two sentences in the Discussion section on deciding on the intervention between RO or OGD to resolve FBO. See from lines 207 to 211.
- We have already highlighted the important changes in the management of FBO to improve EoE in the paragraph between the lines 228 to 237. We have included a sentence in the lines 230 to 232 that changes of management have taken place in our hospital following the results of this study. Is there anything more detailed you suggest us to include?
Comments 5:
Conclusion:
- Make the conclusions more direct and action-oriented (e.g., “ENT services should adopt protocols to ensure all FBO cases undergo appropriate follow-up to exclude EoE”).
Response 5:
- We have included the action between the lines 251 to 256 and such changes have been implemented in the ENT department and the hospital where the study was conducted.
Comments 6:
Comments on the Quality of English Language
The manuscript is well-written and clear. Minor improvements in sentence structure and conciseness are recommended, particularly in the introduction and discussion sections. A language editing service is not required.
Response 6:
If there is anything specific you suggest us addressing the sentence structure and conciseness in the introduction and the discussion sections, we will highly appreciate it.
Many thanks again for your great constructive comments and suggestions for the revision. We really appreciate it.
Round 2
Reviewer 2 Report
Comments and Suggestions for Authors
Dear Authors,
Thanks for the corrections, improvements and revision.
Sincerely